# Evaluation of Fecal Glucocorticoid Metabolite Levels in Response to a Change in Social and Handling Conditions in African Lions (*Panthera leo bleyenberghi*)

**DOI:** 10.3390/ani11071877

**Published:** 2021-06-24

**Authors:** Paula Serres-Corral, Hugo Fernández-Bellon, Pilar Padilla-Solé, Annaïs Carbajal, Manel López-Béjar

**Affiliations:** 1Department of Animal Health and Anatomy, Universitat Autònoma de Barcelona, 08193 Bellaterra, Spain; anais.carbajal@uab.cat; 2Parc Zoològic de Barcelona, Parc de la Ciutadella s/n, 08003 Barcelona, Spain; hfernandez@bsmsa.cat (H.F.-B.); ppadilla@bsmsa.cat (P.P.-S.); 3College of Veterinary Medicine, Western University of Health Sciences, Pomona, CA 91766, USA

**Keywords:** welfare, felids, cortisol, non-invasive, management

## Abstract

**Simple Summary:**

Non-invasive determination of cortisol metabolite concentrations in feces is widely used to evaluate the influence of housing and handling conditions on the stress physiology of wildlife in captivity. The present study aimed to assess the physiological response of a lion pride to a change in management and social conditions after the death of the dominant male of the pride. Before the dominant male died, weekly management routines between the indoor and outdoor enclosures were conducted to avoid cohabitation problems between the two males of the pride. After the death of the dominant male, these weekly management dynamics ceased, leading to a decrease in the daily management routine of the lion pride. An individualized sampling of the animals through the utilization of indigestive markers was conducted, and fecal samples were collected before and after the death of the dominant male. Significant lower cortisol metabolite concentrations in feces were detected after the death of the dominant male, suggesting a positive impact of a decrease in daily management routines, together with a more stable social environment. In addition, assessment of individualized hormone concentrations throughout the study revealed variable physiological responses among lions, providing evidence of the importance of monitoring hormonal profiles individually.

**Abstract:**

Monitoring the hypothalamic–pituitary–adrenal (HPA) axis through determination of fecal cortisol metabolite (FCM) levels is a non-invasive method useful for understanding how handling and social conditions may affect the physiological status of zoo animals. The present study used FCM analysis to evaluate whether the HPA axis activity of a lion pride was modified by a change in social and handling conditions after the death of the dominant male. Five African lions (*Panthera leo bleyenberghi*), two males and three females, were included in the study. Fecal samples were collected before and after the death of the dominant male. To avoid cohabitation conflicts between males before the dominant male died, subgroups were established and subjected to weekly changes between indoor and outdoor facilities. After the death of the dominant male, these management dynamics ceased, and the remaining four lions were kept together outdoors. Significant lower group FCM concentrations (*p* < 0.001) were detected after the decease of the dominant male, probably associated with a decrease in daily handling, together with a more stable social environment. Overall, the present study indicates the effect of different management scenarios on the HPA axis activity and differentiated physiological responses to the same situation between individuals.

## 1. Introduction

In their natural habitat, felid species tend to be solitary animals, except for lions (*Panthera leo*) and, to lesser extent, cheetahs (*Acinonyx jubatus*) [1]. Lions live in groups called prides, which usually comprise four to six consanguineous adult females, their dependent offspring, and one or a few adult males from other prides, thus avoiding crossbreeding between relatives [2,3]. As a group-territorial species, their social structure provides greater defense of their long-term territories and enables communal protection of their offspring [4,5]. Lions have a “fission–fusion” social system, and their pride size varies according to the availability of resources, breaking into subgroups when resources are scarce [6]. Most females remain in the pride when they reach sexual maturity at three years of age, whereas males reach sexual maturity at four years of age and are then evicted from the pride [2]. This male-biased dispersal increases mate choice, as well as benefits kin females to inherit high-quality territories and to remain philopatric [4]. In the wild, females live up to 19 years and males live an average of 16 years, but most of them do not usually exceed 12–13 years of age. On the contrary, in captivity, life expectancy increases to an average of 20 years, but they can live up to 30 years [2,5].

Monitoring welfare is a standard part of zoological practice, even for species such as lions, which are typically inactive over long periods of time throughout the day; in some ex situ conditions they can rest for approximately 10–15 h per day, while in the wild, they can sleep up to 20–21 h per day [7]. To this end, assessment of the hypothalamic–pituitary–adrenal (HPA) axis is a widely used tool, since it reflects physiological changes associated with the stress response, thus providing physiological information on the well-being of the animal [8,9,10]. Yet, an increase in HPA axis activity does not necessarily arise from the response to a stressful stimulus; it can also result from metabolic processes, positive affective states, mating behavior, or physical activity [11,12]. Therefore, behavioral data should be incorporated to contextualize physiological changes when assessing animal welfare.

When a stressor is perceived by the brain, the HPA axis is activated, leading to the release of glucocorticoids (GCs), allowing the organism to cope with the stressful situation and restore homeostasis [13,14]. If the stressor persists and becomes chronic, the physiological response, which is initially adaptive, may eventually turn deleterious, decreasing the animal’s fitness [8,15,16]. Hence, as important stress markers, GC analysis is widely used to assess the index of stress of an animal [14,17,18]. In wildlife, GC determination from blood samples implies capture and/or manipulation, causing added stress that may interfere with the results [14,19]. Alternatively, non-invasive methods such as fecal cortisol metabolite (FCM) determination allow continuous sampling for long periods of time without disturbing the individuals [19,20,21]. There is a time delay between blood circulation of GC and its metabolite excretion in feces corresponding to the intestinal passage time [14]. Thus, FCM values correspond to HPA axis activity hours or days prior to sample collection and represent pooled endocrine activity, a decreasing circadian rhythm effect, and episodic fluctuations [11,22,23]. 

Previous studies have described behavioral and physiological changes in captive felid species subjected to different environmental and social conditions. For instance, a reduced enclosure size has been shown to influence adrenocortical activity in captive Canada lynx (*Lynx canadensis*) [24], as well as to link to abnormal behavior in captive tigers (*Panthera tigris*) and lions [7,25,26]. In four felid species, an environmental disruption because of construction nearby was shown to lead to increased FCM levels, together with altered behavior [27]. Elevated FCM levels have been detected in North American clouded leopards (*Neofelis nebulosa*) kept on exhibit facilities or near potential predators [28]. In primates, a model for social species, the loss of a dominant member may affect their group stability and influence individuals’ stress levels [29]. For instance, in rhesus macaques (*Macaca mulatta*), the removal of a matriarch from her matriline leads to a decreased hierarchical stability and higher levels of cortisol in hair, presumably because of rank changes [30]. In baboons (*Papio hamadryas ursinus*), the death of a close relative results in a significant increase in the GC metabolite levels in their feces [31]. Although changing group memberships in primates has been documented to impact HPA axis activity, the physiological response to a change in social environment stability in lions remains to be evaluated. Additionally, to the best of our knowledge, no specific studies have assessed the physiological response to a decrease in daily management dynamics in captive lions.

Therefore, on account of the unexpected death of a dominant male from a zoo-maintained lion pride, the present study aimed to evaluate if a change in social and handling conditions influenced the HPA axis activity of the lion pride through individualized non-invasive analysis of FCM. We hypothesized that the loss of the group’s dominant member would increase the HPA axis activity of all members of the pride owing to new management conditions and potential changes in hierarchical stability.

## 2. Materials and Methods

### 2.1. Experimental Design

A pride of Southwest African lions (*Panthera leo bleyenberghi*) housed at Barcelona Zoo (Barcelona, Spain, 41°23′16.0″ N 2°11′28.0″ E) was included in this study. The pride comprised five blood-related lions housed together since birth: Three 15-year-old females (identified as F1, F2, and F3), a 15-year-old male (M1), and a three-year-old male (M2).

Fecal samples were collected before and after M1 died due to illness (Figure 1). To avoid cohabitation conflicts between M1 and M2 after several unsuccessful attempts at socializing them before M1 died, two phases were established in which social and management conditions differed: From Monday to Thursday the three females and M2 were kept in the outdoor facilities, while M1 stayed indoor (Phase 1, P1); from Friday to Sunday, and given that M2 displayed anxiety and negative behaviors when left alone indoors, M2 and F1 were kept indoors and M1 had access outdoors, along with F2 and F3 (Phase 2, P2). After the death of M1, these management dynamics were ceased; hence, M2 and the three females were permanently kept together in the outdoor facilities (Phase 3, P3). In the three phases studied, all individuals had indoor access at night.

### 2.2. Housing, Handling, and Environmental Conditions

The five lions were housed in a semi-naturalized exhibit consisting of an inside area of 150 m^2^ out of sight from the visitors and an outside area of 1.090 m^2^ divided into three levels: The upper zone, the cove or medium zone, and the moat or lower zone [32].

The females were treated with contraceptive deslorelin implants to prevent reproduction. The lions were fed once a day early in the morning, from Monday to Saturday, with 3–6 kg of horse or beef bone-in meat. On Sunday, dietary food restrictions were applied, depriving them of food. Water was provided ad libitum. A behavioral enrichment program consisting on stimulation with scents, noises, and objects was conducted daily.

According to historic meteorological data produced by the State Meteorological Agency (AEMET) of the Government of Spain, the average monthly minimum and maximum temperatures throughout the sampling phases ranged from 15.6 to 17.0 °C and from 7.4 to 10.0 °C, respectively, the average monthly relative humidity ranged from 68% to 75%, and the number of days within a month with appreciable precipitations of more than 0.1 mm ranged from two to seven days.

### 2.3. Individual Identification and Sample Collection

As a means to individualize fecal samples in group-housed animals, nontoxic shredded colored waxes (Giotto be-bè^®^, Fila Iberia S.L., Mollet del Vallés, Spain) suitable for children were used as indigestible markers. A different color was assigned to each lion and a piece of meat covered with the shredded wax was given to each individual every morning before feeding.

Fecal samples were collected daily between 8 and 14 h from both the inside and outside facilities prior to cleaning. Feces were collected directly from the ground, avoiding the collection of those in which urine contamination could be observed. Homogenization, to have a representative sample of all defecation, was conducted by collecting different portions of the feces. The collected samples were stored individually in plastic bags and frozen at −20 °C until steroid extraction.

Before hormone extraction, all samples were defrosted and ascribed to each animal according to the wax color observed macroscopically. Of the 147 samples collected, 17 were excluded due to the absence of identifiable colored wax.

### 2.4. Steroid Extraction

Fecal samples were left to dry in an oven (Heraeus model T6; Kendro^®^ Laboratory Products, Langenselbold, Germany) at 60 °C for seven days. Once completely dry, each sample was ground using a Mixer Mill (MM2 type; Retsch^®^, Haan, Germany), obtaining a homogeneous powder of particles of less than 3 mm in size. Hormone metabolites were extracted following a methanol extraction protocol previously described [33,34]. Briefly, 300 mg of the sample powder were weighed on a precision scale and put into a 15 mL conical tube. Afterward, 5.5 mL of 55% methanol (methanol reagent grade 99.9%; Sharlab, S.L., Sentmenat, Spain) was added to every sample and each tube was vortexed (Vortex Mixer S0200-230 V-EU; Labnet International Inc., Edison, NJ, USA) for 30 min. The samples were then centrifuged (Hermle Z300K; Hermle^®^ Labortechnik, Wehingen, Germany) at 1.750× *g* for 15 min and the supernatant was transferred into a micro tube. Immediately after, the fecal extracts were frozen at −20 °C until steroid determination.

### 2.5. Steroid Analysis and Biochemical Validation

For FCM analysis, a commercial enzyme immunoassay (EIA) kit (Cortisol ELISA KIT; Neogen^®^ Corporation, Ayr, U.K.) was used. This EIA kit presents cross-reactivity with prednisolone (47.4%), cortisone (15.7%), 11-deoxycortisol (15.0%), prednisone (7.83%), corticosterone (4.81%), 6β-hydroxycortisol (1.37%), 17-hydroxyprogesterone (1.36%), deoxycorticosterone (0.94%), progesterone (0.06%), and all other steroids (<0.06%).

The EIA was biochemically validated for *Panthera leo bleyenberghi* and the feces matrix by verifying precision (intra- and inter-assay coefficients of variation (CV) from duplicated samples), sensitivity (smallest steroid concentration analyzed), specificity (linearity of dilution), and accuracy (spike-and-recovery test).

Fecal extracts from P1 and P2 were analyzed at the time of sampling; hence, samples collected after M1 had died were assessed with a different EIA kit. Several randomly chosen samples from P1 and P2 were reanalyzed in the EIA kit used for P3, thus serving as a control to verify the variability between assays through inter-assay precision.

### 2.6. Data Analysis

For the longitudinal analysis of individual hormone data, baseline FCM concentrations for each lion before and after the loss of the dominant member were calculated through an iterative process, excluding values greater than the mean plus 1.5 times the standard deviation (SD), until no values exceeded the threshold (mean + 1.5 SD) [35]. Steroid peak concentrations were then considered to be any samples above the aforementioned threshold.

The fecal samples of M1 collected in P1 and P2 (*n* = 16) were excluded from the statistical analysis, since the aim of this study was to evaluate the effect of social and handling changes in the HPA axis for the four individuals that remained alive in the pride.

Statistical analysis of the data obtained was performed using R software for Windows (R-project, Version 3.6.3, R Development Core Team, University of Auckland, New Zealand). The statistically significant level was settled at a *p*-value of <0.05 in all tests performed. Normality of the distribution was assessed by carrying out a Shapiro–Wilk test. Three initial linear mixed-effect models (LMMs) were built to evaluate the effect of the two main factors, the phase (P1, P2, and P3), and the lion (F1, F2, F3, and M2), and their possible interaction on the response variable (FCM) considering the individual as a random effect. Akaike’s information criteria corrected for small sample size (AICc) was used to select the most parsimonious model based upon the delta-AICc and AICc weights. The best-fit model included both main factors with no interaction. Significant differences in mean FCM concentrations where explored with Tukey’s post-hoc tests.

## 3. Results

### 3.1. Biochemical Validation of the EIA

Intra- and inter-assay CV were 9.86 ± 0.16% and 12.31%, respectively, both within the maximum accepted level (see Appendix A). Sensitivity of the EIA kit was 1.94 ng of FCM/g of dry feces. The specificity test showed a correlation between the expected and observed FCM levels of 99.81% (*p* < 0.05). The results of the spike-and-recovery test to measure accuracy presented a mean recovery percentage of 90.07 ± 8.9% (mean ± SD). The results obtained demonstrate that this EIA kit is strongly precise, specific, accurate, and sensible for quantifying FCM concentrations.

### 3.2. Mean FCM Concentrations

The levels of FCM were detected in all samples analyzed. The sample size and mean (±SD) concentrations of FCM per group (F1, F2, F3, and M2; M1 excluded) and the individuals within each phase are provided in Appendix A. The heterogeneous sample size among the individuals and phases was due to the fact that color could not be identified in several collected samples (*n* = 17), together with the interindividual variation in defecation patterns.

Summary statistics for the FCM data are shown in Table 1. Significant differences in the group FCM concentrations among phases were detected (*p* < 0.001). The post-hoc test revealed that the FCM concentrations differed between P1 and P3 (*p* < 0.001) and between P2 and P3 (*p* < 0.05) (Figure 2). Between P1 and P2, the FCM levels did not differ significantly (*p* > 0.05). Regarding individual mean FCM concentrations, no significant differences were detected between the individuals (*p* > 0.05).

### 3.3. Longitudinal Assessment of Individual Hormonal Profiles

The individual baseline mean, baseline cut-off, peak mean FCM concentrations, and proportion of peaks before and after the death of the dominant male are summarized in Table 2. The baseline mean (±SD) FCM concentrations ranged from 14.25 ± 3.73 to 19.28 ± 5.49 ng/g and from 10.90 ± 2.24 to 17.76 ± 3.93 ng/g, before and after the death of the dominant male, respectively.

All samples collected and subsequently analyzed to determine the FCM concentrations are presented chronologically in separate graphs for each lion (Figure 3). On eight occasions in which more than one sample was collected per day and individual, the mean FCM levels are presented in the graphs.

A decrease in baseline FCM levels along with a reduction in the proportion of FCM peaks and baseline cut-off amplitude was observed in the sub-adult male (M2) after the dominant male died. A similar pattern was observed in F1 and F3, where, even though more variability was seen in the FCM levels compared to M2 after the death of the dominant male, a decline in baseline FCM concentrations, as well as a lower amplitude of the baseline cut-off, were noticeable. The hormonal changes observed in M2, F3, and F1 could not be identified in F2, in which the baseline FCM concentrations remained mostly constant across the study.

## 4. Discussion

This study examined whether a change in management and social conditions influenced the HPA axis activity in captive lions. Contrary to expectations, the levels of FCM decreased after the decease of the dominant male. The reduced adrenocortical activity detected could indicate a reduction in stress-inducing factors.

Non-invasive methods to assess adrenocortical activity must be rigorously validated for the species investigated prior to any assay [23,36,37]. In addition to the biochemical validation of the assay, performing a biological and physiological validation is also recommended [23,37]. These procedures require further animal management that may not always be possible or desirable, especially in wildlife or zoo-kept animals, as in the present study. Although biological and physiological validations could not be performed, previous studies suggest that fluctuations in FCM reflect reliable biological changes in the family Felidae [24,27,38]. However, given that GCs are also released in response to situations not normally regarded as stressful when assessing welfare, the monitorization of adrenocortical activity should be accompanied by other biological or behavioral measures to contextualize and correctly interpret the hormonal changes observed [8,39].

In the present study, in order to avoid direct contact between both males of the pride, weekly management was carried out and subsequently stopped once the dominant male had died. The results revealed lower FCM levels after the decease of the dominant male probably associated with the decreased management of the animals, together with the new social situation. In four other felid species (*Panthera pardus*, *Leptailurus serval*, *Panthera pardus saxicolor*, and *Uncia uncia*), the addition of a disruption to their environment due to construction works in a nearby enclosure was shown to lead to an increase in FCM concentrations [27]. In our study, when the environmental disruption caused by increased daily management routines and social instability ceased, the mean group FCM concentrations decreased. Therefore, in accordance with Chosy and collaborators [27], our findings suggest that a more stable social and environmental situation may be associated with a reduction in adrenocortical activity. Before the death of the dominant male, the weekly management consisted of moving several individuals from indoor to outdoor facilities; thus, they spent three to four days per week in each facility. A previous study in captive lions demonstrated that restrictive environments, such as indoor facilities, were less stimulating and individuals presented altered day- and night-time behavior compared to animals kept in exhibit areas, reflecting an unsatisfied motivation for activity and, consequently, the presence of stress [7]. Contrary to Kohari and collaborators [7], no significant differences in FCM levels were detected between the two phases in which this weekly management was carried out. Even though our results did not reveal an influence of restrictive environments in the individuals’ HPA axis, the limited sample size precludes from making a valid conclusion.

Factors related to social structure, such as the presence of conspecifics and aggressive interactions, can elicit physiological responses [40,41]. An important limitation of the present study is the absence of objective behavioral data complementary to physiological assessment. Therefore, although taken cautiously, behavioral information regarding social dynamics obtained from zoo keepers’ observations was considered. According to the zookeepers’ observations, F1 was the dominant lion when placed outdoors with F2, F3, and M2 (P1) and M1 was the dominant lion when kept outdoors with F2 and F3 (P2). After the death of M1 (P3), F1 remained the dominant lion. Lions, like hyenas, have fission–fusion structures that may allow conflict management by avoiding encounters with dominant members [42]. Notwithstanding that, before the dominant male died, the two males demonstrated aggressive behavior when indirectly exposed to each other, M2 was beginning to exhibit hierarchical behavior and F3, at times, was seen trying to expel him. All of the interactions observed among the individuals studied could be interpreted as the presence of an unstable hierarchy, given that when a hierarchy evolves from instability to a stable situation, aggressive encounters should progressively decrease [40]. However, in the absence of objective behavioral information, the aforementioned results should be interpreted with caution. According to the literature reviewed, stress and, therefore, elevated GC levels, can occur either in high- or low-rank individuals depending on the social organization of the species or the population [29,41]. In our study, no significant differences in FCM levels were identified between individuals; thus, no significant relationship between FCM levels and social status was found. Still, the results should be interpreted with caution due to the small sample size.

In addition to the generalized decrease in FCM levels, the changes observed in the hormonal profile of M2 after the dominant male died could be interpreted as decreased HPA axis activity due to a lack of sensorial exposure to M1, given that indirect exposure to an aggressive member is sufficient to elicit a physiological response [28].

Regarding F2, her FCM concentrations remained mostly constant throughout the study; thus, F2 was probably not influenced by the change in social and management conditions. This may illustrate the variability of each individual to perceive factors as stressors [15,28,40]. Before the dominant male died, F2 remained outdoors the whole week; thus, this lioness was already subjected to less management. Nonetheless, F3 was subjected to the same social and handling conditions as F2 throughout the study, yet longitudinal evaluation of the hormonal profile of F3 suggested a decrease in baseline FCM levels and a lower amplitude of the baseline cut-off after the dominant male died. This contrasting result demonstrates how challenging it can be to assign inter-individual FCM variation to a single factor [43].

The small study group precluded an evaluation of sex and age as factors. Although in African lions no differences in FCM concentrations among age groups have been observed [44,45], there is no consensus regarding the influence of sex on FCM levels in felids [28,46]. Thus, in future studies, these variables should be considered.

## 5. Conclusions

Through the evaluation of FCM levels, we expected to see an increase in HPA axis activity after the death of a pride’s dominant member related to a change in social and handling conditions. Contrary to expectations, the group FCM concentrations significantly decreased after the death of the dominant male, revealing a reduction in adrenocortical activity potentially associated with the decrease in daily management routines, together with a more stable social environment in the pride.

The present study evidenced how FCM determination is an effective non-invasive method for the long-term evaluation of adrenocortical activity in zoo-kept animals and, thus, a useful tool to assess and improve their welfare. Furthermore, our results corroborate the positive impact of decreased daily handling and social stability on HPA axis activity while demonstrating how variable physiological responses may occur between individuals exposed to the same situation. These findings highlight the importance of monitoring FCM concentrations individually rather than as a group sample when assessing animal welfare, since variable physiological responses may occur among individuals.

## Figures and Tables

**Figure 1 animals-11-01877-f001:**
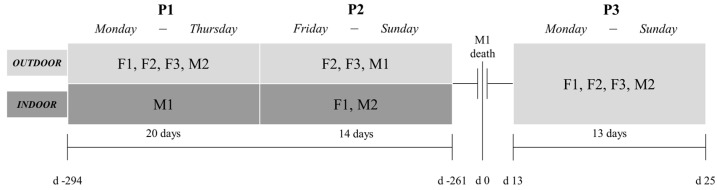
Experimental design of the study. Before the death of M1, weekly management, including two phases (**P1** and **P2**), was carried out to avoid cohabitation conflicts between M1 and M2. After the death of M1, these management dynamics were ceased (**P3**). A total of 20 days were sampled for P1, 14 days for P2, and 13 days for P3. Days from the death of M1 are indicated for each phase. Abbreviations: F1, female 1; F2, female 2; F3, female 3; M1, male 1; M2, male 2.

**Figure 2 animals-11-01877-f002:**
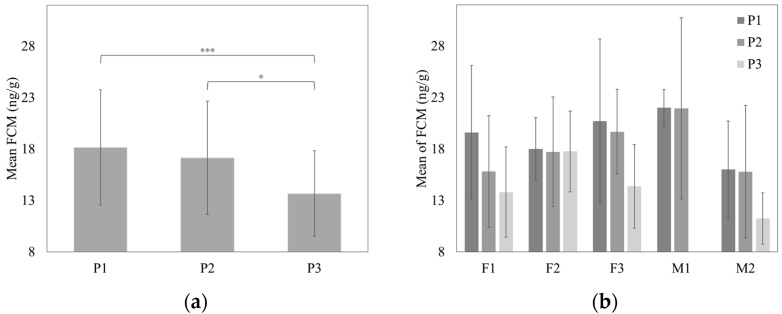
Plot of means (±SD) of fecal cortisol metabolite (FCM) concentrations before (P1 and P2) and after (P3) the death of the dominant male showing: (**a**) Group (F1, F2, F3, and M2) means; (**b**) individual means. Asterisk indicates * *p* < 0.05 and *** *p* < 0.001. Abbreviations: F1, female 1; F2, female 2; F3, female 3; M1, male 1; M2, male 2.

**Figure 3 animals-11-01877-f003:**
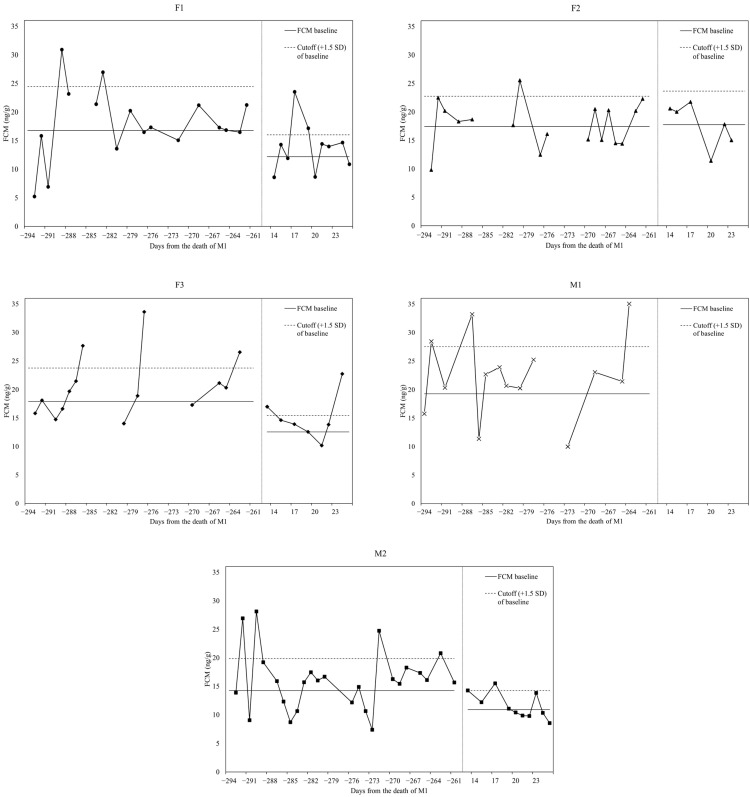
Longitudinal representation of fecal cortisol metabolite (FCM (ng/g)) levels before and after the death of the dominant male (**M1**) in: (**F1**) Female 1; (**F2**) Female 2; (**F3**) Female 3; (**M1**) Male 1; (**M2**) Male 2. The vertical dashed line symbolizes a jump in time from the sampling finalization before the death of M1 and sampling resumed afterward. The discontinuity of the line observed between several samples reflects those days where no wax color could be detected in the samples collected; thus, such fecal samples could not be identified or ascribed to an individual.

**Table 1 animals-11-01877-t001:** Statistical results for the effect of phase and lion on the FCM concentrations, showing model selection based on Akaike’s information criteria corrected for small sample size and the best fit linear mixed model output. Selected model is denoted in bold.

Model Selection									
**Model**		**K**		**AICc**		**Δ** **AICc**		**Weight**	
**FCM~Phase + Lion**		**8**		**700.00**		**0.00**		**0.63**	
FCM~Phase		5		701.09		1.09		0.36	
FCM~Phase ∗ Lion		4		709.14		9.14		0.01	
**Final Model Output**									
**Factor**	**Estimate**		**Standard Error**		***t*****-Value** (**df**)		***p-*** **Value**		**95% CI**
Intercept	18.47		1.87		9.88 (108)		<0.001		[14.80, 22.13]
Phase (P2)	−1.13		1.15		−0.99 (108)		0.33		[−3.38, 1.12]
Phase (P3)	−4.48		1.11		−4.03 (108)		<0.001		[−6.66, −2.30]
Lion (F2)	0.86		2.54		0.34 (108)		0.74		[−4.13, 5.84]
Lion (F3)	1.64		2.55		0.64 (108)		0.52		[−3.37, 6.64]
Lion (M2)	−2.29		2.46		−0.93 (108)		0.36		[−7.11, 2.54]

Abbreviations: K, number of parameters of the model; AICc, Akaike’s information criteria corrected for small sample size; ΔAICc, difference between the model AICc score and AICc of the lowest model; df, degrees of freedom; CI, confidence interval.

**Table 2 animals-11-01877-t002:** Individual baseline (±SD) fecal cortisol metabolite (FCM) concentrations, baseline cut-off (calculated as baseline levels + 1.5 ∗ SD), peak mean (±SD) FCM concentrations, and proportion of peaks before and after the death of the dominant male of the lion pride.

Individual	Time	FCM (ng/g)			Proportion of Peaks
		Baseline	Baseline Cut-Off	Peak Mean	
Female 1 (F1)	Before	16.78 ± 5.12	24.45	28.93 ± 2.81	11%
	After	12.17 ± 2.55	16.00	20.35 ± 4.51	20%
Female 2 (F2)	Before	17.46 ± 3.53	22.75	25.53 ^1^	6%
	After	17.76 ± 3.93	23.66	-	0%
Female 3 (F3)	Before	17.89 ± 3.90	23.74	29.25 ± 3.80	20%
	After	12.55 ± 1.95	15.48	19.85 ± 4.08	25%
Male 1 (M1)	Before	19.28 ± 5.49	27.51	33.55 ± 1.33	19%
	After	-	-	-	-
Male 2 (M2)	Before	14.25 ± 3.73	19.84	25.15 ± 3.22	15%
	After	10.90 ± 2.24	14.25	15.50 ^1^	8%

^1^ In those cases where only one peak was detected within a time, the peak mean (±SD) could not be calculated, thus the value of the peak is shown in the table.

## Data Availability

The data presented in this study are available on request from the corresponding author.

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
