# Peer review of "Evaluation of Fecal Glucocorticoid Metabolite Levels in Response to a Change in Social and Handling Conditions in African Lions (Panthera leo bleyenberghi)"

_animals, 2021, doi:10.3390/ani11071877_

Round 1
Reviewer 1 Report
This study describes the effects of changes in social composition and husbandry procedures on FCM levels in a small pride of zoo housed lions. The topic is important as it relates to the management and welfare of captive lions, but unfortunately, the study is flawed in its design, the sample size is very low, and important measures are missing (e.g., activity levels, intra-group social relationships, group dynamics before and after M1’s death). The results are missing important information as well (see below), and the conclusions are not supported by the data.
I also had concerns regarding the assumption that cortisol levels serve as a good indicator of stress (in particular, when no other measurement is done to provide a context). Cortisol is a better indicator of arousal levels, which can be either good (e.g., in response to food) or bad (e.g., avoiding a predator).
Finally, while the change in husbandry routines was driven by the death of the male, these are two distinct factors (i.e., the death and the change in husbandry). However, as a reader I felt confused at times since they appear to be used interchangeably (e.g., Abstract: The present study used FCM analysis to evaluate whether the HPA axis activity of a lion pride was modified by a change in social and handling conditions after the death of the dominant male. Introduction: Although changing group memberships in primates have been documented to impact the HPA axis activity, the physiological effect of the loss of a group member in lions remains to be evaluated.
General questions:
What were the types and quality of relationships in the group before and after M1’s death? These are likely to have an effect on FCM levels.
How long have the animals been housed together? How are they related to each other?
Specific questions:
Introduction:
In general, I found the Introduction well written and organized.
55 – please change “of” to “off”.
63-65 – please explain how a measure of the HPA axis provides information about the “general physiological status of animals”.
Methods:
Why was M1 housed alone indoors whereas M2 was housed with F1?
What was the cause of death of the male?
What was the trigger to begin collecting these samples? Were the animals adjusted to it?
127 – how often were the females treated with deslorelin? Are there any effects of deslorelin on corticosteroids secretion?
2.3. Individual identification and sample collection – were the samples checked for possible urine contamination?
150 – Ref 30 relates to extraction from hair cortisol in cattle. Please provide a reference for fecal samples in lions (e.g., https://doi.org/10.1093/conphys/cot021).
173 – are you referring to the levels measured in p3? It is somewhat confusing to refer to those as “baseline” measurements.
210-211 – please clarify - Regarding the evaluation of mean FCM levels among lions, no significant differences were detected (p > 0.05).
220 – please change the order to: and from 10.90 ± 2.24 ng/g to 17.76 ± 3.93 ng/g.
Table 2 – please specify that the Baseline cut-off levels are calculated as baseline levels + 1.5*SD.
228-230 – please explain why you chose to include samples collected throughout the day and not at specific time frames in light of the known circadian effect of corticosteroids secretion. Since there are 7 occurrences of averaged samples, this can skew the data.
Results:
There are additional data that you need to report in your results (e.g., fixed effects estimates, confidence limits, and the models with the AIC scores).
With the current presentation of the results, it is impossible to tease apart the individual contribution of M1’s death vs. change in husbandry procedures on FCM levels. How do you determine if the change in levels is due to a loss of a pride member, resulting from less handling, or both?
Discussion
248-250 – I find this conclusion to be unsupported by your data. There are additional factors that can influence levels of cortisol that have not been accounted for and are not associated with the perception of stress (e.g., physical activity). In addition, the FCM peaks that you mention appear to have occurred at least 2 weeks after M1 died (with the exception of F3), which, again, doesn’t support your conclusion.
256-258 – but there are also many other examples of others that reject the reliability of FCM fluctuations as a sole reliable indicator of stress. Please add this view as well.
266 – what indications do you have that the social environment became more stable after M1’s death?
267-268 – this needs to be rephrased. A reduction in FCM levels does not lead to a reduction in adrenocortical activity, it results from it.
268 – I understand what you were attempting to say here, but it needs to be changed since it’s not exactly similar to your study. In the referenced studies, the disruption was added and, in your study, it was removed. Not the same. Also, when exposed to chronic stressors, the HPA activity changes from hyper to hypo activity. If the animals were chronically stressed by this procedure, how does that influence your conclusions?
277-280 – I suggest toning this conclusion down as the sample size is very small.
What are the effects and significance of such changes in social vs. solitary animals?
294-296 – this statement is false.
Reviewer 2 Report
Comment to the paper: Evaluation of fecal glucocorticoid metabolite levels in response to a change in social and handling conditions in African lions (Panthera leo bleyenberghi) by Paula Serres-Corral , Hugo Fernández-Bellon , Pilar Padilla-Solé, Annaïs Carbajal, Manel López-Béjar
In this study, authors investigate the possible changes in HPA activity of a lion pride at Barcelona zoo, in relation to different social and handling conditions, consisting of two different subgrouping of individuals for two different periods of the week, for 294 days, due to incompatibility between the two male members. Evaluation was made by means of fecal corticosterone metabolites (FMC) monitoring. As main result, after the death of the elder male, and after the reunion of the overall individuals in one group, and, thus, the reduced management routines, the levels of individual FMC significantly decreased. The authors stress the role of FCM determination as non-invasive method for long term evaluation of adrenocortical activity in zoo-kept animals, and thus as useful tool to assess and improve their welfare.
The topic is interesting, and this paper might contribute to strengthen the usefulness of information on FMC levels for various purposes linked to animal welfare. However, considering even the very small sample size, to judge reliable the observed FMC reduction we must be sure that this is not due to bias depending on sample processing. Therefore, some crucial aspects of the research paradigm need to be clarified, to make the paper publishable.
Here my comments.
Line 104 -116
Some information is missing, for example the causes of the death of the male M1. Was it due to some illness?
In which period of the year did phases 1-2 and phase 3 take place? Considering the time elapsed between phases 1-2 and phase 3, other factors may be related to the differences found, such as different seasonal weather conditions, e.g. outdoor ambient temperature. If phase 1-2 occurred in winter, perhaps cold temperatures could have stimulated the HPA axis for thermoregulation more than in phase 3. The opposite condition would strengthen the results, at least on this aspect. Thus, weather data (high and low mean temperatures, rainfall, humidity) should be reported for each sampling period and differences should be evaluated.
Line 146
Even if not included in the statistical analysis it would have been useful to show the data for phase 1-2 regarding the other male, M1. Have his samples been determined?
Line 150-152
The protocol for steroid extraction that authors cited in the methods refers to dairy cows. The authors left the lion fecal samples drying for seven days instead of the 48-h indicated in that protocol. Authors should report if they made a stability hormone test to understand the effect of high temperature for that long period on hormone metabolite concentrations.
Line 192
Authors show similar intra- and inter-assay CV levels when fecal samples were assessed utilizing the Neogen cortisol EIA kit. Those levels seem slightly high, especially the intra-assay CV. There are a lot of steroid hormone EIA kits that could be used for fecal sample assessments with even lower intra- and inter-assay CV. I believe that it could be useful to know the mean FCM levels of the samples used for these tests. In fact, the knowledge of these data could be relevant for making the decision to use this specific EIA kit for FCM measurements, especially if it is possible to indicate a concentration range in which the assessment variability is lower. Did the authors verify if most of the samples had concentrations similar to the 50% B/B0 of the standard curve and if all were inside the range 20-80 % b/b0, where most of the standard curves are linear?
Further, authors must clarify if they used the same kit lot number for all the analyses, or samples from phase 3 were assessed at a different time with a different kit lot number. Did authors have control samples to verify the variability among different assessments?
Line 281-328
Effect of rank. A hierarchical order is constructed through the statistical processing of ethological observations on agonistic behaviour, such as threats, submissions and displacements. In the absence of objective data, the authors can take social dynamics into account but they should be more sparing in discussing results they don't have.
Reviewer 3 Report
The article is very good and original. I suggest small and few changes in the text, noted in the pdf copy that I attached. My suggestions do not diminish my appreciation that the article is highly publishable.

Author Response
Response to Reviewer 3 Comments
The article is very good and original. I suggest small and few changes in the text, noted in the pdf copy that I attached. My suggestions do not diminish my appreciation that the article is highly publishable.
Response: We thank the reviewer for these comments.
Point 1: Figure 2b is redundant since this data is the same of Table1. Delete it.
Response 1: The reviewer is right to point out that Figure 2b presents the same data of Table 1. Following the reviewer’s suggestion and having into account a comment given by another reviewer, we have modified the results section by moving Table 1 from the manuscript to Supplementary Materials Table S1.
Improvements in the MS:
Line 217: The sentence has been modified to: Sample size and mean (± SD) concentrations of FCM per group (F1, F2, F3 and M2; M1 is excluded) and individual within each phase are provided in Supplementary Materials Table S1.
Line 230: The paragraph has been modified to: Summary statistics for FCM data are shown in Table 1. Significant differences in group FCM concentrations among phases were detected (p < 0.001). The post hoc test revealed that FCM concentrations differed between P1 and P3 (p < 0.001) and between P2 and P3 (p < 0.05) (Figure 2).
Point 2: Line 263-265 – The clouded leopard findings do not appear to be equivalent to the findings in the present article with lions. The stress induced in leopards is related to the presence of people (more zookeepers and more visitors), and with being closer to potential predators. These sources of distress are quite different from the handling of enclosures and the presence of a dominant individual in the study reported here. So while it may "look like social stress", the situation is very different. Later in the text, the authors are right to discuss the results with this clear focus on hierarchical instability and changing rooms.
Response 2: We thank the reviewer for pointing this out. The reviewer is correct in indicating that the sources of distress reported in the clouded leopard study are different from the ones reported here. Accordingly, we have excluded the reference and modified this paragraph of the discussion.
Improvements in the MS:
Line 287: The revised text reads as follows: Results revealed lower FCM levels after the decease of the dominant male probably associated with the decreased management of the animals together with the new social situation. In four other felid species (Panthera pardus, Leptailurus serval, Panthera pardus saxicolor and Uncia uncia), the addition of a disruption to their environment due to construction works in a nearby enclosure led to an increase in FCM concentrations [26]. In our study, the removal of an environmental and social disturbance from the lion pride, resulted in a decrease in FCM concentrations. Therefore, in accordance with Chosy and collaborators (2014), our findings suggest that a more stable social and environmental situation may be associated with a reduction in adrenocortical activity.
Point 3: Line 271 – “studied here” It is wordiness.
Response 3: We have removed the words “studied here” on line 296. Moreover, in agreement with the reviewer’s suggestion, we have also removed the word “studied” on line 285.
Point 4: Line 298-303 – The sentence is speculative and unimportant for the present work.
Response 4: We agree with the reviewer that the sentence is speculative and not relevant for the results presented in the study. Accordingly, we have removed this sentence from the manuscript.
Point 5: Line 304-313 – The sentence is unimportant for the interpretation of results.
Response 5: We agree with the reviewer that the sentence is not relevant for the interpretation of results. Therefore, as suggested by the reviewer, we have modified this paragraph of the discussion.
Improvements in the MS:
Line 329: This sentence has been modified from the discussion section and now reads as follows: In addition to the generalized decrease of FCM levels, the changes observed in the hormonal profile of M2 after the dominant male died could be interpreted as a decreased HPA axis activity due to lack of sensorial exposure to M1, given that indirect exposure to an aggressive member is sufficient to elicit a physiological response [25].
Point 6: Line 321 –change “lion” for lioness
Response 6: Corrected.
Round 2
Reviewer 2 Report
The authors improved their manuscript, however some issue has still to be addressed.
Abstract
In the end authors state: “Overall, the present study documents the effect of different management scenarios on the HPA axis activity and differentiated physiological responses to the same situation between individuals.”
Considering the small sample of subjects, the possible effects due to sex differences in corticosteroid metabolism, and the low efficiency of the used EIA kit, authors should replace “documents” with “indicates” or “suggests”.
Response 2 - Climatic data
Line 141: The following sentence has been included: According to the data produced by the State Meteorological Agency (AEMET) of the Government of Spain, average monthly minimum and maximum temperatures throughout the sampling phases ranged from 15.6 to 17.0 °C and 7.4 to 10.0 °C, respectively; average monthly relative humidity ranged from 68 to 75 %; and number of days within a month with appreciable precipitations of more than 0.1 mm ranged from 2 to 7 days.
Authors should clearly state if these data are referred to the real sampling time or are referred to historical averages.
Response 4
“homogenization prior to steroid extraction requires taking different portions of the feces in order to have a representative sample of all defecation, increasing the sample volume and thus prolonging the drying process.”
Authors should report also in the paper this information.
Author Response
The authors improved their manuscript, however some issue has still to be addressed.
Abstract
In the end authors state: “Overall, the present study documents the effect of different management scenarios on the HPA axis activity and differentiated physiological responses to the same situation between individuals.”
Considering the small sample of subjects, the possible effects due to sex differences in corticosteroid metabolism, and the low efficiency of the used EIA kit, authors should replace “documents” with “indicates” or “suggests”.
Response: Following the reviewer’s suggestion, we have modified the sentence on line 39 to: Overall, the present study indicates the effect of different management scenarios on the HPA axis activity and differentiated physiological responses to the same situation between individuals.
Response 2 - Climatic data
Line 141: The following sentence has been included: According to the data produced by the State Meteorological Agency (AEMET) of the Government of Spain, average monthly minimum and maximum temperatures throughout the sampling phases ranged from 15.6 to 17.0 °C and 7.4 to 10.0 °C, respectively; average monthly relative humidity ranged from 68 to 75 %; and number of days within a month with appreciable precipitations of more than 0.1 mm ranged from 2 to 7 days.
Authors should clearly state if these data are referred to the real sampling time or are referred to historical averages.
Response: In agreement with the reviewer, we have emphasized that meteorological data refers to historical averages. The sentence on line 141 has been modified to: According to historic meteorological data produced by the State Meteorological Agency (AEMET) of the Government of Spain, average monthly minimum and maximum temperatures throughout the sampling phases ranged from (…).
Response 4
“homogenization prior to steroid extraction requires taking different portions of the feces in order to have a representative sample of all defecation, increasing the sample volume and thus prolonging the drying process.”
Authors should report also in the paper this information.
Response: In accordance with the reviewer suggestion, the following sentence has been incorporated on line 155: Homogenization to have a representative sample of all defecation was conducted by collecting different portions of the feces.